# Identification of Modifiable Risk Factors of Exacerbations Chronic Respiratory Diseases with Airways Obstruction, in Vietnam

**DOI:** 10.3390/ijerph191711088

**Published:** 2022-09-04

**Authors:** Thuy Chau Nguyen, Hoa Vi Thi Tran, Thanh Hiep Nguyen, Duc Chien Vo, Isabelle Godin, Olivier Michel

**Affiliations:** 1Department of Family Medicine, Pham Ngoc Thach University of Medicine, Ho Chi Minh City 740500, Vietnam; 2Nguyen Tri Phuong Hospital, Ho Chi Minh City 740500, Vietnam; 3School of Public Health, Université Libre de Bruxelles (ULB), 1050 Brussels, Belgium; 4Clinic of Immuno-Allergology, CHU Brugmann, Université Libre de Bruxelles (ULB), 4 Place A-Van Gehuchten, 1020 Brussels, Belgium

**Keywords:** asthma, chronic obstructive pulmonary disease, occupational exposure, indoor ventilation

## Abstract

Objectives: to determine modifiable risk factors of exacerbations in chronic respiratory diseases with airways obstruction (i.e., asthma and COPD) in southern Vietnam. Methods: an environmental and health-related behavioural questionnaire was submitted to patients with both chronic respiratory symptoms and airways obstruction. An exacerbation was defined as any acute worsening in clinical symptoms requiring a change in treatment, in a patient receiving prophylactic therapy. Results: 235 patients were evaluated, including 131 (56%) chronic obstructive pulmonary disease (COPD) and 104 (44%) asthmatics. There were 75% males and 69% smokers. Occupational exposure accounted for 66%, mainly among construction and industry workers. Smoking was associated with more severe airways obstruction. Respiratory exacerbations were reported in 56/235 patients (24%). The risk of exacerbation was increased in patients with a lower education level, exposure to occupational pollutants, cumulative smoking ≥ 20 pack year, housing space < 10 m^2^, and poorly ventilated housing. Based on multivariate analysis, the risk of exacerbation remained significantly higher among patients with occupational exposure and low housing space per person. Conclusions: besides smoking cessation, more supportive policies, including improvement of occupational environment and housing design for better ventilation, are needed to prevent the severity of chronic respiratory diseases in Vietnam.

## 1. Introduction

Asthma and chronic obstructive pulmonary disease (COPD) are the two most common chronic respiratory diseases (CRD) with airways obstruction, in the world and they have a very serious impacts on public health [1]. These diseases are clinically characterized by both chronic respiratory symptoms and exacerbations, the last defined as an acute worsening of symptoms that results in additional therapy. Consistently, the GOLD guidelines for the treatment of COPD recommend taking into consideration the patient’s symptoms and risk of exacerbation for the pharmacological management of the disease [2]. Without optimal treatment and preventive strategies for exacerbations, COPD often progresses to severe grade with persistent symptoms, worsening airways obstruction defect, frequent admissions to hospital, and increasing mortality [3]. Consistently, there is a relationship between accelerated lung function decline and exacerbations of COPD [4]. Among asthmatics, recently, a longitudinal study has also demonstrated that exacerbations are associated with a decline in lung function [5].

The prevention of exacerbations needs a pharmacological and a non-pharmacological approach. According to the World Health Organization (WHO), modifiable risk factors are strongly associated with increased mortality in asthma and COPD. These factors include tobacco smoking habits, indoor air pollution from biofuels, urban outdoor air pollution, occupational factors, and exposure to respiratory allergens and/or ambient particulate matter [6]. Based on data from the Global Burden of Diseases, Injuries, and Risk Factors Study (GBD) 2017, in South-East Asia, besides smoking, household air pollution from solid fuels and occupational risks are the highest risk factors for chronic respiratory disease-attributable DALY (disability-adjusted life-year) rates [7]. Vietnam is a low medium-income country with rapid economic growth and a clear difference in population density between rural and urban areas. In Vietnam, asthma and COPD treatment was mainly based on a pharmacological intervention and smoking cessation. There is still a great need for non-pharmacological measures in the prevention of exacerbation, such as reducing occupational exposure and improving housing conditions. This study aims to determine what are the modifiable exacerbation risk factors of asthma and COPD in an urban population of southern Vietnam.

## 2. Material and Methods

### 2.1. Population Selection

From November 2020 to May 2021, 235 patients older than 18 years with at least one chronic respiratory symptom (cough, sputum production, wheezing, chest tightness, dyspnea) and obstructive defect on spirometry were successively evaluated at Nguyen Tri Phuong Hospital (Vietnam) using a cross-sectional study design.

Patients with active tuberculosis, cancer, acute infectious disease, unstable cardiovascular disease, were excluded by anamnesis, physical examination, chest radiograph, and 12-lead electrocardiogram. Systemic diseases, autoimmune diseases, and HIV infection were ruled out by physical examination and further testing when risks and symptoms were suspected.

Baseline spirometry was measured one month after exacerbation, and only patients with airflow obstruction were invited to participate in the study and were formally enrolled in the study after signing the informed consent form. Patients were checked for use and inhalation technique when performing the bronchodilator (BD) test. Afterwards, each patient was invited to complete a questionnaire.

The study protocol was approved by the Ethics Committee of Pham Ngoc Thach University, Ho Chi Minh City, Vietnam (008/HDDD) and was enrolled in the ClinicalTrials.gov database (NCT04232579).

### 2.2. Pulmonary Function Test

The Forced Expiratory Volume in 1 s (FEV1, in L) and Forced Vital Capacity (FVC, in L) were measured using a spirometer (Medisoft Ltd., Sorinnes, Belgium), after stopping bronchodilators for >12 h. Measurements were obtained by technicians, trained according to European Respiratory Society guidelines [8].

Airflow obstructive limitation was defined as FEV1/FVC < LLN (Lower Limit of Normality—according to Global Lung Initiative for Southeast Asians) [9].

A post-bronchodilator (BD) (400 mcg inhaled salbutamol) increase of ≥200 mL and ≥12% in FEV1 compared with baseline values was defined as significant.

### 2.3. Questionnaire

Each patient completed a questionnaire (see Appendix A) including demographic and socio-economic information, occupation, medical history, type of housing, indoor pollution factors and behavioural risks for chronic respiratory diseases. Each patient was asked to identify his/her type of accommodation based on pictures and defined as: apartments had one of a set of large rooms in a building with some rooms inside, windows and a main entrance, old apartments were built ≥20 years; tube houses were narrow, long and built close together, can be built with several floors; rent houses were small ground floor rooms with poor ventilation, frequently with no flush toilets and a kitchen in the main living area; rural houses were separated houses built of wood or semi-solid materials found in suburban districts with no flushed toilets.

### 2.4. Definitions

COPD was diagnosed in the presence of a non-significant post-BD FEV1 associated with smoking and/or cooking with biofuels [2]. Asthma diagnosis was based on reversible post-BD FEV1, whatever the environmental exposure or irreversible post-BD FEV1 when patients were not exposed to tobacco smoke or cooking with biofuels [10]. Combined COPD and asthma were defined as Chronic Respiratory Diseases (CRD) with airways obstruction. Exacerbation was defined as any acute worsening in clinical symptoms compared to the baseline, requiring a change in medication treatment, in a patient receiving prophylactic therapy [2,10]. Severe exacerbation was defined as an acute episode with worsening of baseline symptoms, leading to hospitalization or intensive care unit (ICU) admission. The degree of dyspnea was determined according to the modified Medical Research Council (mMRC) scale of 0 to 4 [2,10]. Patients were assessed as having knowledge about chronic lung disease when answering “Yes” to more than 2 questions of the questionnaire (Appendix A).

Smoking levels were assessed using the pack-year index, which is calculated by multiplying the number of packs of cigarettes smoked per day by the number of years of smoking. Exposed occupations to dust, fumes, and chemicals included industrial worker, construction worker, driver, farmer and others with particulate matter exposure [1,6]. The housing space per person was defined as the mean floor area (in square metres) of a housing unit divided by the average household size.

### 2.5. Data Analysis

IBM SPSS Statistics version 23.0 (IBM Co., Armonk, NY, USA) was used for data analysis. Mean values were presented with 95% confidence intervals (95% CI). The χ^2^ test was used for univariate analysis to test differences in the risk factors among all patient characteristics. The continuous data were compared using student *t*-test. Logistic regression analysis was used to identify predictors of exacerbation occurrence. A multivariate model was applied to each parameter significant in univariate analysis. Odds Ratios (OR) and 95% CI were calculated and *p* ≤ 0.05 was defined significant.

## 3. Results

### 3.1. Demographic and Socio-Economic Characteristics of Study Patients

Among the 235 patients, 175 (74.5%) were male and the mean age was 61.5 years (60.1–62.9). The age at onset was ≥40 years for 96.2%. There were 76 patients (32.3%) who migrated from the rural provinces to the urban area of HCMC.

Looking at educational history, 40 patients (17.0%) did not receive any scholarly education, 87 patients (37.0%) reached primary school, 91 patients (38.7%) secondary and 17 patients (7.2%) post-secondary level. Exposed occupation accounted for 154 patients (65.5%), mainly for industry workers 52 (22.1%), construction workers 25 (10.6%), car/trucks drivers 31 (13.2%), 23 farmers (9.8%) and other occupation exposure 23 (9.8%).

Exacerbations were observed in 56 (23.8%) patients, with 32/56 (57.1%) being severe exacerbations. Among them, 45/56 (80.4%) had occupational exposure of which 18 (40.0%) were construction or industry workers, 12 (26.7%) drivers, 7 (15.6%) farmers, and 8 (17.8%) in other occupations. There were 14.9% (23/154) severe exacerbations among exposed workers, compared to 11.1% (9/81) among non-exposed workers. There were 172/235 (73%) patients with low income (average monthly salary < 5 million VND-equivalent to USD 200/month). Only 76/235 (32.3%) of them have general knowledge about their own respiratory disease. A higher risk of exacerbation was associated with occupational exposure to pollutants and to a low level of education (Table 1).

### 3.2. Medical History and Clinical Characteristics of the Studied Population (Table 1)

Medical history was characterised by tuberculosis (13.6%), hypertension (31.9%), and type 2 diabetes (11.1%). Family history of asthma and COPD were observed among 24.3% and 4%, respectively. The mean body mass index (BMI) was 22.3 (21.8–22.8), and the baseline SpO_2_ value was 97.0% (96.1–97.8), of which SpO_2_ < 95% was recorded in 13 cases (5.5%).

They were 131 (55.7%) COPD patients and 104 (44.3%) asthmatics. Compared to asthma, COPD was a risk factor for exacerbation, though not statistically significant. The degree of dyspnea according to mMRC was, respectively, 47.2%, 40.0%, 8.5% and 4.3% for grades 1, 2, 3, and 4 and no patients had mMRC grade 0. A higher risk of exacerbation was associated with a mMRC degree ≥ 2 and FEV1 post-BD < 60% PV.

Allergic history (rhinitis, sinusitis, conjunctivitis, eczema, urticaria, swelling) was present in 40.4%, but not related to exacerbations.

Compared with non-smokers, smokers had more severe lung function defects, characterized by a post-bronchodilator FEV1/FVC [66.2 (64.3–68.1) % vs. 58.3 (56.8–59.7) %; *p* < 0.001] and FEV1 [62.0 (57.7–66.2) % of predicted value, vs. 56.2 (53.3–59.0) %, *p* < 0.05].

### 3.3. Smoking Habits and Smoking Exposure of the Studied Population (Table 1)

There were 162/235 (68.9%) smokers, in which 159/162 (98.1%) were male, 60/162 (37.0%) current smokers and 102/162 (63.0%) ex-smokers. Cumulative smoking ≥ 10 and ≥20 pack-years accounted for 144/235 (61.3%) and 124/235 (52.8%), respectively. Among the smokers, 79% and 21% were COPD patients and asthmatics, respectively.

Exacerbation occurred in 27.2% of current smokers compared to 16.4% of non-smokers corresponding to an approximately two-fold increased risk factor and was significantly more frequent in patients with cumulative smoking ≥ 20 pack years. The prevalence of a history of parental (especially paternal) tobacco exposure in childhood was 45.5%, though not associated with an increased risk of exacerbation.

### 3.4. Types of House and Indoor Pollution Factors (Table 2)

There were 41.7% of patients living in tube houses, 31.5% in non-tube houses, 8.9% in rental houses, 7.2% in old apartments (built ≥ 20 years), 1.7% in new apartments (built < 20 years), and 8.9% in rural houses. While most of the patients lived on the lower floors (<2 floors), only 0.4 % lived on the upper floors (≥7 floors). The average house space per person was 19.5 m^2^ (16.8–22.2) of floor/person, of which space ≤ 10 m^2^ accounted for 39.6%. Housing space per person below the national average in Vietnam (defined as ≤ 25 m^2^ of floor/person) accounted for 79.1%; 70.6% of houses did not have extractor fans, 60.0% had no windows, and 57.0% used air conditioners. Indoor burning incense accounted for 81.7%; 14.5% of patients were exposed to fumes from biofuels, 85.5% used gas and electric stoves as all patients lived in urban Ho Chi Minh City; 36.6% were exposed to volatile compounds, and 29.8% to insecticides. Respectively, 29.4%, 23.0%, and 16.2% had domestic pets, cockroaches, and rats at home. There were 6.0% of patients living within a distance of <100 m near the public entertainment centre, 4.3% near an industrial park, 2.6% near a bus/coach station, and 1.7% near landfill.

**Table 2 ijerph-19-11088-t002:** Types of house and indoor pollution factors and the risk of exacerbations.

Characteristics	n (%)	Exacerbation + n (%)	OR (95% CI)	*p* Value *
Total	235	56		
Type of house				
Others	116 (49.4)	34 (29.3)	1	0.052
Tube and rent house	119 (50.6)	22 (18.5)	0.55 (0.29–1.01)	
Housing space per person				
<10 m^2^ of floor/person	93 (39.6)	29 (31.2)	1	0.032
≥10 m^2^ of floor/person	142 (60.4)	27 (19.0)	0.52 (0.29–0.95)	
Extractor fan				
No	166 (70.6)	46 (22.7)	1	0.030
Yes	69 (29.4)	10 (14.5)	0.44 (0.21–0.94)	
Air conditioner				
No	101 (43.0)	28 (27.7)	1	NS
Yes	134 (57.0)	28 (20.9)	0.69 (0.38–1.26)	
Indoor incense burning				
No	43 (18.3)	13 (30.2)	1	
Yes	192 (81.7)	43 (22.4)	0.67 (0.32–1.39)	NS
Using biofuel for stove				
No	201 (85.5)	45 (22.4)	1	NS
Yes	34 (14.5)	11 (32.4)	1.66 (0.75–3.66)	
Using volatile compounds				
No	149 (63.4)	40 (26.8)	1	NS
Yes	86 (36.6)	16 (18.6)	0.62 (0.32–1.20)	
Using insecticides				
No	165 (70.2)	42 (25.5)	1	NS
Yes	70 (29.8)	14 (20.0)	0.73 (0.37–1.45)	
Childhood biofuel exposure				
No	49 (20.9)	8 (17.8)	1	NS
Yes	186 (79.1)	48 (25.3)	1.57 (0.68–3.59)	

* χ^2^ test; OR = Odds Ratio; CI = Confidence Interval.

Good ventilation in houses using an air extractor and housing space per person ≥ 10 m^2^ were associated with less frequent occurrence of exacerbations (Table 2). Exacerbations were more frequent in patients with exposure to biofuels (32.4%) but the difference was not significant. The prevalence of a history of parental biofuel exposure in childhood was 79.1%, though not associated with an increased risk of exacerbation (Table 2).

### 3.5. Risk Factors for Exacerbation Occurrence on Univariate Analysis

Detailed data are given in Table 1 (demographic, education, occupation, clinical status, spirometry, and smoking habits) and Table 2 (types of house and indoor pollution factors).

The significant modifiable risk factors are summarized in Figure 1. An exposed occupation and low level of education were associated with a greater risk of exacerbation occurrence, whereas good ventilation in houses using air extractors and housing space per person ≥ 10 m^2^ were protective against exacerbation occurrence. Exacerbations were significantly more frequent in patients with cumulative smoking ≥ 20 pack years.

### 3.6. Risk Factors for Occurrence of Exacerbation on Multivariate Analysis (Table 3 and Figure 2)

We developed a multivariate model adjusted for age and sex which was applied to each parameter significant in univariate analysis. The risk of exacerbation remained significantly higher among patients with occupational exposure and low housing space per person ≤ 10 m^2^ (Table 3).

**Table 3 ijerph-19-11088-t003:** The risk of exacerbations after adjusted multivariate analysis for gender and age, of each factor significant on univariate analysis. Only significant adjusted factors are shown below.

	Adjusted OR (95% CI)	*p* Value *
Occupational exposure		
No	1	
Yes	2.46 (1.17–5.21)	0.018
Housing space per person		
≤10 m^2^	1	0.046
>10 m^2^	0.53 (0.28–0.99)	

* Significance at 5% level; OR = Odds Ratio; CI = Confidence Interval.

The risk of exacerbation was significantly higher among smokers with occupational exposure or housing space per person ≤ 10 m^2^ than in non-smokers (Figure 2).

**Figure 2 ijerph-19-11088-f002:**
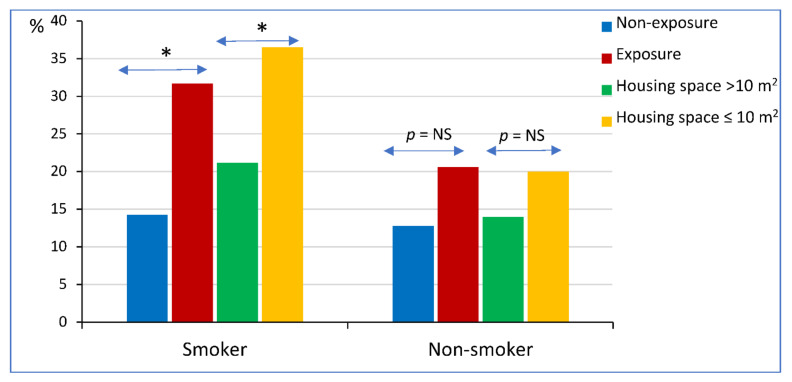
Comparison of the risk of exacerbations in relation to occupational exposure and housing space, among smokers and non-smokers patients. * χ^2^ test, *p* ≤ 0.05; NS: Non Significative.

## 4. Discussion

### 4.1. Smoking

Smoking is the main respiratory risk of CRD, especially for COPD [7]. Cigarette smoking is also a risk factor for promoting irreversible airway obstruction in asthma patients [11] according to our data (smokers among 79% of COPD patients and 21% asthmatics).

The pathogenic effects of tobacco smoke are cumulative with the dose and duration of use [11]. The exacerbation risk is significantly higher in smokers (COPD [12] and asthmatics [13]). The relationship between cumulative smoking with the exacerbation rate remains questionable, though a defect in lung function is more frequent in patients with cumulative smoking ≥ 10 pack-years [14,15]. In our study, on one side, smokers were at higher risk of exacerbation than non-smokers, particularly if cumulative smoking was ≥ 20 pack years, and, on the other side, compared with non-smokers, smokers had more severe lung function defects. In Vietnam, specialized units for smoking cessations are not yet covered by health insurance. Providing insurance coverage for treatment with medications and counseling to vulnerable populations may have a significant impact on smoking cessation. More effective health education about the harmful effects of tobacco is needed, especially for young people and low-income populations.

Childhood exposure to passive tobacco smoke may be also a risk factor for exacerbation in adulthood with an obstructive syndrome, especially in maternal tobacco smoke [16]. This risk factor is probably insignificant in Vietnam though because the female smoking rate is only 1% [17]. In this study, 46% of patients were exposed to paternal tobacco smoke during childhood, though they were not associated with an increased risk of exacerbation at the adult age.

### 4.2. Occupational Exposure

Occupational exposure is one of the primary causes of respiratory symptoms and lung function impairment [1]. Many studies showed that occupational exposure is characterised by regular exposure to fumes, inorganic and organic fine dust, and many chemicals [18,19,20]. The industry, construction, and agriculture workers are the main occupations, associated with the risk of CRD [21,22,23]. In the present study, the most commonly exposed occupation among the patients were industry workers (22.1%), construction workers (10.6%), and drivers of vehicles using petrol (13.2%). Due to the characteristics of the job in the urban area of Ho Chi Minh City, farmers accounted for only 9.8% of patients.

In The Netherlands, occupational exposure to vapors, gases, dusts, fumes (VGDF), and pesticides is associated with a decreased level of lung function and a high prevalence of airway obstruction in the general population [24]. There were significant associations between occupational exposure to VGDF and chronic airflow limitation, COPD, and emphysema [25]. A high incidence of COPD was also found to be associated with occupational exposure to VGDF [26,27]. In a population with high employment in the textile industry, chronic bronchitis symptoms and airflow obstruction were associated with occupational exposures to VGDF and lung function defect was related to the duration of occupational exposure [28]. Coal miners, hard-rock miners, tunnel workers, concrete-manufacturing workers, and non-mining industrial workers have been shown to be at the highest risk for developing COPD [29]. This association was also described among dairy farmers for asthma and COPD [22]. Workplace exposures contribute to the burden of multiple chronic respiratory diseases, including asthma and COPD. The proportion of deaths due to avoidable occupation exposure was estimated at 16% for asthma and 14% for COPD [30].

A higher risk of exacerbation was associated with occupational exposure to asthma and COPD in many studies. Exposure to gas, smoke, organic dust, dampness and mould, cold conditions, and jobs handling low molecular weight agents (isocyanates, aldehydes, anhydrides, colophony, dyes…) were associated with exacerbation of asthma [19]. The risk of acute exacerbations and the level of severity of COPD were associated with exposure to biomass, organic dust, and mineral agents [27].

An interaction between smoking and occupation was reported in other studies. The combined effect of smoking and occupational exposure to dusts, gases or fumes was an independent risk factor for a high incidence of COPD [31]. Combined exposure to both smoking and occupational factors markedly increased the risk of COPD [32]. There was an increased smoking-adjusted risk for developing COPD in men exposed to respirable crystalline silica, gypsum and insulation material, diesel exhaust, and high levels of particles from asphalt/bitumen and welding fumes. In the same study, an increased risk was also observed among women highly exposed to various organic particles from soil, leather, plastic, soot, animal, textiles, and flour [33].

Our study suggests a combined effect of smoking and occupation on the risk of exacerbations of CRD. To prevent these airways diseases in Vietnam, spirometries should be included in the medical screening organised by occupational medicine. Currently, this systematic screening is not yet implemented in most factories, due to a lack of medical education to perform validated measurements but also, due to the difficulties for the patients to consent to perform spirometries. Therefore, associated with smoking cessation, there is a need for policy interventions to reduce occupational exposure levels, with the aim to control the burden of CRD with airways obstruction, especially in developing countries.

### 4.3. Indoor Air Pollution

According to WHO, almost seven million premature deaths in the world were attributable to the joint effects of ambient and household air pollution [34]. The deterioration of indoor air quality is a combined result of poor ventilation and harmful pollutants produced by biofuel, volatile organic compounds, particulate matter, aerosol, and others [35]. In optimal conditions to ensure good ventilation, living space must be large enough and must guarantee sufficient space to meet the privacy needs of the occupants; the average housing space per person must be 37 m^2^ of floor/person [36].

In our study, 40% had housing space per person ≤ 10 m^2^. Compared with the national average housing space per person in Vietnam (which was defined as 25 m^2^ of floor/person) [37], 79% of our patients did not meet the recommendations. In addition, 71% of houses did not have extractor fans, 60% had no windows, while the indoor burning incense habit accounted for 82%. Among Buddhist residents in Asian countries, burning incense is a common source of indoor particulate matter. Various categories of respiratory ailments including COPD and asthma were associated with the frequency of burning incense, the room size and incense volume, and the number of persons in a room [38]. An asthma-screening program on adolescents showed a negative association between lung function and daily exposure to incense burning, sharing a bedroom, and living in a house adjacent to a traffic road [39]. The harmful effects on the respiratory system of incense smoke were also found with chronic obstructive pulmonary disease [40]. In our study, incense burning, volatile compounds, and insecticides were not risk factors for exacerbations; however, with the study being cross-sectional, we can not exclude that the more severe patients have improved the indoor air of their dwelling.

The impact of indoor air pollution on asthma and COPD exacerbations risk and mortality has been reported, especially in Asian populations [41,42,43]. Similar to occupational exposure, there was a possible synergy between indoor air pollution and tobacco exposure in exacerbation risk [44]. In the present study, the risk of exacerbation remained significantly higher among smokers with low housing space per person ≤ 10 m^2^. There was a possible combined effect between smoking and housing space on the occurrence of exacerbation in COPD patients.

Besides the quality of indoor ventilation, biomass exposure has also been described to be an indoor pollutant and may adversely affect the progression of CRD, particularly in developing countries [45]. Cooking with biofuels has been demonstrated to be a risk factor for female and non-smoker COPD patients [2]. Biomass exposure has a similar impact on exacerbation risk when compared with tobacco smoke [46]. In our study, an exacerbation was more frequent in patients with biofuel exposure but the difference was not significant. Few patients were using biomass energy for cooking (14.5%), and all of them were living in an urban area.

Therefore, in addition to smoking cessation, it is necessary to propose to the Ministry of Construction in Vietnam to issue regulations to ensure ventilation in housing construction.

### 4.4. Strengths and Limitations of the Study

This cross-sectional study analyses the relationship between different modifiable factors and exacerbations risk. An overestimation when calculating ORs was probable. Therefore the causal effects of occupation and house ventilation, on the risk of CRD exacerbations, need to be confirmed using a longitudinal study design. Also, the limited sample size could be not representative of all urban populations of southern Vietnam, and therefore, repeated studies with larger populations could provide more robust evidence for the association between modifiable risk factors and disease severity.

Moreover, there is a possible recall bias when answering the questionnaire that may lead to an underestimation of the occurrence of exacerbations because the patient may self-medicate. Exploiting information through the retrospective questionnaire may give inaccurate results.

The study was conducted in a general hospital which was not a centre dedicated to only respiratory diseases in Ho Chi Minh City. Indeed, since Nguyen Tri Phuong Hospital takes care of various patients with many specialties, the indications for a lung function test are not always considered and convincing patients to consent to spirometry is also difficult.

## 5. Conclusions

In Vietnam, smoking, exposure to occupational pollutants, and limited housing space per person have associated risk factors for exacerbations of CRD with airways obstruction (i.e., COPD and asthmatics).

The primary prevention of exacerbations in those patients should take into consideration the respiratory modifiable risk factors. This intervention needs, not only individual awareness and education of the patient but also support from health and social policies, such as promotion of smoking cessation, improvement of the occupational environment to protect workers, and legislation concerning housing design for better ventilation.

## Figures and Tables

**Figure 1 ijerph-19-11088-f001:**
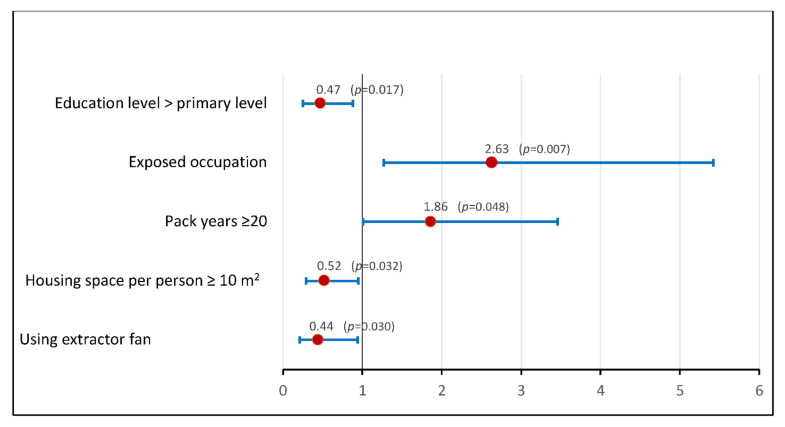
Modifiable factors related to exacerbation in asthma and chronic obstructive pulmonary disease. Horizontal bars indicate 95 percent confidence interval. Odds ratio is shown on the marker located on the horizontal bars.

**Table 1 ijerph-19-11088-t001:** The risk of exacerbations in regard to demographics, education and occupation, clinical status, spirometry and smoking habits.

Characteristics	n (%)	Exacerbation + n (%)	OR (95% CI)	*p* Value *
Total	235	56		
Sex				
Female	60 (25.5)	13 (21.7)	1	NS
Male	175 (74.5)	43 (24.6)	1.18 (0.58–2.38)	
Education level				
Primary	127 (54.0)	38 (29.9)	1	0.017
Secondary/Post-secondary	108 (46.0)	18 (16.7)	0.47 (0.25–0.88)	
Occupation				
Non-exposed	81 (34.5)	11 (13.6)	1	0.007
Exposed	154 (65.5)	45 (29.2)	2.63 (1.27–5.42)	
History of tuberculosis				
No	203 (86.4)	45 (22.2)	1	NS
Yes	32 (13.6)	11 (34.4)	1.84 (0.82–4.10)	
History of allergy				
No	140 (59.6)	32 (22.9)	1	NS
Yes	95 (40.4)	24 (25.3)	1.14 (0.62–2.10)	
Clinical diagnosis				
Asthma	104 (44.3)	19 (18.3)	1	NS
COPD	131 (55.7)	37 (28.2)	1.76 (0.94–3.29)	
mMRC grade				
grade 1	111 (47.2)	17 (15.3)	1	0.005
grade ≥ 2	124 (52.8)	39 (31.5)	2.48 (1.31–4.71)	
FEV1 post-BD				
<60% PV	128 (54.5)	37 (28.9)	1	0.046
≥60% PV	107 (45.5)	19 (17.8)	0.53 (0.28–0.99)	
Smoking status				
No	73 (31.2)	12 (16.4)	1	0.074
Yes	162 (68.8)	44 (27.2)	1.90 (0.93–3.85)	
Pack years ≥ 20				
No	111 (47.2)	20 (18.0)	1	0.048
Yes	124 (52.8)	36 (29.0)	1.86 (1.01–3.46)	
Exposure to smoke from parents in childhood				
No	128 (54.5)	26 (21.5)	1	NS
Yes	107 (45.5)	30 (26.3)	1.30 (0.71–2.38)	

*** χ^2^ test; OR = Odds Ratio; CI = Confidence Interval; COPD = Chronic Obstructive Pulmonary Disease; m MRC = modified Medical Research Council; FEV1 = Forced Expiratory Volume in one sec; BD = bronchodilator; PV = Predicted Value.

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
