# Peer review of "Identification of Modifiable Risk Factors of Exacerbations Chronic Respiratory Diseases with Airways Obstruction, in Vietnam"

_ijerph, 2022, doi:10.3390/ijerph191711088_

Round 1

Reviewer 1 Report

Thank you for asking me to review this paper. Important results were found.

1. I suggest revising the paper using a writing guide such as Stroebe guidelines such that important information is not missed. For example no where in the methods is the design mentioned. Some how you can think it is cross sectional but extensive data on exacerbations is presented

2. The term chronic obstructive respiratory disease (CORD) is  not standards I suggest they use asthma and COPD. 

3. The title has nothing about exacerbations, mentions risk factors of obstructive disease but in the paper it appears the paper is about exacerbations. More information about exacerbations is needed in the introduction

4. The results of the factors associated are impressive but how homogenous asthma and COPD patients are in this analysis is not clear.

5. There are so many tables, some with just 2 rows. These should be reduced and those with similar content combined. 

6. Table on univariate analysis can be left out

7. I did not see any figure, consider illustrating some of the results with  figures

8. There will be need to revise the English by review by an English speaking author.

Author Response

Response to Reviewer 1  

  1. I suggest revising the paper using a writing guide such as Stroebe guidelines such that important information is not missed. For example no where in the methods is the design mentioned. Some how you can think it is cross sectional but extensive data on exacerbations is presented.

Response: the design (“cross sectional study”) has been included in the methods.

  1. The term chronic obstructive respiratory disease (CORD) is not standards I suggest they use asthma and COPD. 

Response: done

We have replaced CORD by CRD (chronic respiratory diseases) with airways obstruction (i.e. COPD and asthma).

  1. The title has nothing about exacerbations, mentions risk factors of obstructive disease but in the paper it appears the paper is about exacerbations. More information about exacerbations is needed in the introduction

Response:

  • title has been adapted “Identification of modifiable risk factors of exacerbations among chronic respiratory diseases with airways obstruction, in Vietnam”
  • a paragraph with additional information on “exacerbation in asthma and COPD” has been added in the introduction

(“These diseases are clinically characterized by both chronic respiratory symptoms and exacerbations, the last defined as an acute worsening of symptoms that results in additional therapy. Consistently, the GOLD guidelines for treatment of COPD recommends to take into consideration the symptoms and risk of exacerbation for their pharmacological management (2). Without optimal treatment and preventive strategies for exacerbations, COPD often progress to severe grade with persistent symptoms, worsening airways obstruction defect, frequent admissions to hospital and increasing mortality (3). Consistently, there is a relationship between accelerated lung function decline and exacerbations of COPD (4). Among asthmatics, recently, a longitudinal study has also demonstrated that exacerbations are associated with a decline in lung function (5).”)

  1. The results of the factors associated are impressive but how homogenous asthma and COPD patients are in this analysis is not clear. (?)

Response: 

In table 1 we shown that the relative prevalence of asthma and COPD were comparable and that the risk factor of exacerbations was slightly higher for COPD, though not significantly different compared to asthmatics. Therefore the populations were homogeneous in regard with the risk of exacerbations.

  1. There are so many tables, some with just 2 rows. These should be reduced and those with similar content combined. 

Response: tables 1, 2 and 4 have been combined and table 3 suppressed (data into the text)

Tables 6 and 7 have been replaced by figures 2 and 3

  1. Table on univariate analysis can be left out:

Response: tables 1, 2, 4 on univariate analysis were combined to simplify the presentation of the data.

  1. I did not see any figure, consider illustrating some of the results with figures

Response:  we agree that figure are more comprehensive compared to table. We added 2 figures.

  1. There will be need to revise the English by review by an English speaking author.

Response:  has been reviewed by an English speaking people

Reviewer 2 Report

Chau et al. studied the effect of several modifiable risk factors on exacerbations of CORD (including COPD and asthma) in a mainly urban population in Vietnam. Especially occupational exposure and low housing space per person were found to associate with increased risk of exacerbations, and possible modification of these factors by smoking was observed.

Major comments:

1.     The study setting is cross-sectional, and the outcome (exacerbations) is relatively common (23.8%), thus the ORs are likely to overestimate the actual effect to some extent. Have you considered using other estimation methods, for example Poisson regression, that would allow more accurate estimation of the risk with unadjusted and adjusted prevalence ratios (PR) instead of ORs?

2.     Page 5, smoking habits: In the analysis smoking was categorized into no and yes, and it seems that the ex-smokers were included in the yes category. However, ex-smokers may differ from the current smokers. Have you analyzed this as a three-category variable (no, ex, current) and would that affect the results? Numbers of current smokers and ex-smokers would seem to permit analyzing them as separate categories.

3.     Page 5, parental smoking: How many participants reported unclear childhood tobacco smoke exposure (page 13, questionnaire, IV. 1.5.) and in which category were they included in the analysis?

4.     Page 6 Table 5: The estimates of the effects of incense burning, volatile compounds, and insecticides are on the protective side (though NS). Due to the cross-sectional nature of the study, this could be due to avoidance of using compounds that cause respiratory symptoms. It would be good to a little bit discuss this possibility in section 4.3. “Indoor air pollution”, chapter 4 that covers incense burning. In addition, the general limitations due to cross-sectional setting should be considered in the section 4.4. “Strengths and limitations”.

5.     Page 7, Table 6: It would be interesting to see the results for all the included variables, even if they are not significant.

6.     Page 7. Table 7. The results here imply of possible combined effect between smoking and occupational exposure and housing space. Have you considered additionally analyzing this with interaction term(s) in a regression (logistic or Poisson) model?

Minor comments:

1.     Abstract: page 1, row 10: I think that “Background” does not describe the contents of the first chapter here, you could consider using e.g. “Objectives” instead.

2.     Page 2, row 83: Old apartment is here defined as >10 years, but elsewhere a definition of >20 years is used.

Author Response

Response to Reviewer 2

Major comments:

  1. The study setting is cross-sectional, and the outcome (exacerbations) is relatively common (23.8%), thus the ORs are likely to overestimate the actual effect to some extent.

Have you considered using other estimation methods, for example Poisson regression, that would allow more accurate estimation of the risk with unadjusted and adjusted prevalence ratios (PR) instead of ORs?

Response: the study design is effectively cross-sectional and only OR can be estimated by the model. The scientific literature does document an overestimation given by OR. POR could indeed be used, the procedure for obtaining them is much the same. This study aims essentially to highlight the risk factors associated with exacerbation; it is more of a descriptive than an explanatory study

  1. Page 5, smoking habits: In the analysis smoking was categorized into no and yes, and it seems that the ex-smokers were included in the yes category. However, ex-smokers may differ from the current smokers. Have you analyzed this as a three-category variable (no, ex, current) and would that affect the results? Numbers of current smokers and ex-smokers would seem to permit analyzing them as separate categories.

Response: when analyzing in separate categories:  

         - there were no significative difference between current smoker (11/ 44 =25%) and ex-smokers (33/44 = 75%), p= 0.053

         - there were no significative difference between non - smoker (12/ 23 =52.2%) and current smokers (11/23 = 47.8%), p= 0.774

         - the difference between non smoker (12/45 = 26.7%) and ex-smoker (33/45 = 73.3%) was significative with p= 0.018

Therefore, we combined smoker + ex-smokers.

  1. Page 5, parental smoking: How many participants reported unclear childhood tobacco smoke exposure (page 13, questionnaire, IV. 1.5.) and in which category were they included in the analysis?

Response: The responses of all participants were No or Yes. In fact, no participant have reported unclear.

  1. Page 6 Table 5: The estimates of the effects of incense burning, volatile compounds, and insecticides are on the protective side (though NS). Due to the cross-sectional nature of the study, this could be due to avoidance of using compounds that cause respiratory symptoms. It would be good to a little bit discuss this possibility in section 4.3. “Indoor air pollution”, chapter 4 that covers incense burning.

Response: done

In addition, the general limitations due to cross-sectional setting should be considered in the section 4.4. “Strengths and limitations”.

Response: done

  1. Page 7, Table 6: It would be interesting to see the results for all the included variables, even if they are not significant.

Response: Table 6 was renamed as table 3. The results for all variables (even if they are not significant) were described in the new tables 1 and 2

  1. Page 7. Table 7. The results here imply of possible combined effect between smoking and occupational exposure and housing space. Have you considered additionally analyzing this with interaction term(s) in a regression (logistic or Poisson) model?

Response : Table 7 was replaced by figure 3.

The adequacy of the model was tested by the Hosmer-Lemeshow.

Minor comments:

  1. Abstract: page 1, row 10: I think that “Background” does not describe the contents of the first chapter here, you could consider using e.g. “Objectives” instead.

Response: done

  1. Page 2, row 83: Old apartment is here defined as >10 years, but elsewhere a definition of >20 years is used.

Response: done

Round 2

Reviewer 1 Report

No further comments